# Using the COAsT Python Package to Develop a Standardised Validation Workflow for Ocean Physics Models

David Byrne[1], Jeff Polton[1], Enda O' Dea[2], and Joanne Williams[1]

[1]National Oceanography Centre, Liverpool, UK
[2]Met Office, Exeter, UK

**Correspondence:** Jeff Polton (jelt@noc.ac.uk)

**Abstract.** Validation is one of the most important stages of a model's development. By comparing outputs to observations, we can estimate how well the model is able to simulate reality, which is the ultimate aim of many models. During development, validation may be iterated upon to improve the model simulation and compare to similar existing models or perhaps previous versions of the same configuration. As models become more complex, data storage requirements increase and analyses improve, scientific communities must be able to develop standardised validation workflows for efficient and accurate analyses with an ultimate goal of a complete, automated validation.

We describe how the COAsT Python package has been used to develop a standardised and partially automated validation system. This is discussed alongside five principles which are fundamental for our system: system scaleability, independence from data source, reproducible workflows, expandable code base and objective scoring. We also describe the current version of our own validation workflow and discuss how it adheres to the above principles. COAsT provides a set of standardised oceanographic data objects ideal for representing both modelled and observed data. We use the package to compare two model configurations of the Northwest European Shelf to observations from tide gauge and profiles.

# 1  Introduction

15  Numerical modelling plays a vital part in both the prediction and understanding of ocean processes. Models must be validated in order to determine their accuracy. Generally, this is done by comparing their output to analogous observed data or data sets derived from observations with the aim of quantifying how close model simulations are to the reality they attempt to replicate. The reason for this is clear: operational forecasting models must be accurate for the communities receiving predictions, and scientific simulations must be able to adequately represent the processes we seek to study.

20  New and existing model configurations must go through a development process during which input data sets are defined and refined, physical parameterizations are chosen and their parameter values are tuned. The repeated validation required during this iterative process is one of the most important stages of model development. However, it can often be time and resource consuming, lack consistency and performed in a new and arbitrary fashion for subsequent model configurations.

The modelling community benefits from a multitude of pre-existing packages across various programming languages, which 25 serve to assist in model validation. The Python ecosystem, in particular, offers data manipulation packages such as numpy, pandas, xarray, and Dask (McKinney et al., 2010; Hoyer and Hamman, 2017; Rocklin, 2015), which commonly serve as the building blocks to a wide variety of data analysis frameworks, although they are not specific to oceanographic modelling. More specialized Python packages are available which offer a variety of metrics and diagnostics, including UTide (Codiga, 2011), RapidMOC (Roberts, 2017), PyWavelets (Lee et al., 2019), python-gsw (Firing et al., 2021), and python-seawater (CSIRO). 30 Additionally, there are packages such as PyFVCOM that are geared towards analyzing the output of specific numerical models. While many of these packages belong to broader suites or collections, such as SEAPY or METRIC (Castruccio, 2021), there is currently no universal Python framework to standardize oceanographic data structures across varying model configurations and packages.

In this paper, we introduce and demonstrate the Coastal Ocean Assessment Toolbox (COAsT) Python package, which offers 35 a framework for standardising oceanographic data structures into which the validation and analysis of numerical models can be integrated. We describe the relevant components of COAsT alongside our philosophy and key principles behind the development of the package as well as our validation workflows. Some of these ideas are demonstrated as we set out our workflow for validation of temperature and salinity again ocean profiles and sea surface height against tide gauge data.

In the following sections we describe some of the decisions made in the initialisation of our validation workflow. In the 40 remainder of this section we provide more information on the COAsT package alongside details on the model and observed data used in this paper. In Sections 2 – 4, we showcase some of the analyses available within the package and present results for tidal data, non-tidal residuals and tracers (temperature and salinity) respectively. Finally, in Section-5, we discuss the future of this workflow and opportunities for its expansion.

## 1.1  The COAsT Framework

COAsT is a Python package and framework that aims to standardise many of the aspects of the analysis of modelled and observed oceanographic datasets. At its core, the package provides the user with a set of standardised data classes which are

designed for oceanographic analyses. By using these standardised structures, a user can be sure that analysis code will work for their data, regardless of source, so long as it adheres to the appropriate structure. The package builds upon key libraries including Xarray (Hoyer and Hamman, 2017) and Dask (Rocklin, 2015). The dataset contained within each instance of a

50 COAsT data class is an Xarray dataset, which is a structured and labelled multidimensional array. Depending on the class in question, the names, structure and layout of dimensions and variables are enforced and controlled by a .json file, which can optionally be modified.

As an example, the Gridded class is designed to be used with model data on a regular grid and must contain an xarray Dataset that has dimensions *t_dim* (time), *z_dim* (vertical dimension), *y_dim* and *x_dim* (horizontal dimensions). Typically, this kind

of data would come from the output of a numerical model. At the time of writing, the package has been tested and used with output from the NEMO model, although this could be extended to other models in the future. The data should be stored in any xarray compatible file (e.g. netCDF, zarr). Once data has been ingested into the COAsT framework, the same validation script can be more easily applied repeatedly as dimension names and variable names will be known and any necessary preprocessing steps will have been taken.

Now that we have loaded our model data into the framework, suppose we wish to apply an analysis to temperature data taken from vertical profiles of the ocean, e.g. observations from CTD casts. As discussed later in the paper, COAsT provides a *Profile* data class for this purpose. Each instance of a *Profile* object can contain a dataset representing multiple profiles. This dataset must adhere to a predefined structure and naming conventions, specifically an index dimension called *id_dim* and a vertical dimension called *z_dim*. If a user has profile data from different sources, then they can be easily combined within the

COAsT system. The Gridded class is used to read, represent and manipulate output from two NEMO model runs, and its use interactively with the Profile and Tidegauge classes allows a comparison between the model and observed data.

The COAsT package uses an object-oriented approach, and classes can be broadly separated into two types: data and analysis classes. Data classes are those discussed above, wherein the structured datasets are read, stored and manipulated. Data classes are an ideal place to keep manipulation or visualization routines, e.g. subsetting data based on some criteria or plotting some

70 known variable, as the class is aware of the structure of the dataset it contains. Analysis classes are more flexible and general, with no real constraints. They contain code for analysis of one or multiple data objects. (For example, analysis classes include *GriddedStratification, ProfileAnalysis, TidegaugeAnalysis, Transect* and *Contour*). Alternatively, analyses can be contained within external scripts or libraries, which make use of the fundamental COAsT structures.

At the time of writing, the data classes have two main parent classes, from which all others are derived: the *Gridded* class and

75 *Indexed* class. These are summarised in Table-1. The *Gridded* class enforces a data structure ideal for output from structured models such as NEMO[1]. The *Indexed* class is a general class for data that can lie along a single index, such as the *Profile* data discussed earlier. It has a generalised dimension structure with an index dimension *id_dim*, a time dimension *t_dim* and a vertical dimension *z_dim*. The children of this class have different permutations of these dimensions, and they are summarised in Table-2.

---

[1]https://www.nemo-ocean.eu (last accessed 28 June 2022)

**Table 1.** The two main data classes in the COAsT package and their function. The *Gridded* class is used in this paper to represent output data from the NEMO model. Child classes of the Indexed class are used to represent observed data (see Table-2)

| Class | Data Structure | Function |
|---|---|---|
| **Gridded** | – **Dimensions:** (t_dim, z_dim, y_dim, x_dim)<br>– **Coordinates:** time, depth, longitude, latitude | For representing gridded data such as from a structured model or reanalysis dataset. Contains additional routines for reading some file formats and manipulating data in space and time. |
| **Indexed** | – **Dimensions:** (id_dim, z_dim, t_dim)<br>– **Coordinates:** time, longitude, latitude, depth | For representing data that can lie along a single index. Can be used for many observation types and has many subclasses for different instruments such as *Profile*, *Tidegauge* and *Glider*. These additionally contain routines for reading from some common observation databases as well as manipulation of datasets in time and space. |

**Table 2.** Four examples of COAsT class which are children of the Indexed class. These are the classes used to represent observed data. Other classes are available within the COAsT package.

| Class | Data Structure | Function |
|---|---|---|
| **Profile** | – **Dimensions:** (id_dim, z_dim)<br>– **Coordinates:** id, latitude, longitude, depth, time | For representing one or more vertical oceanographic profiles. Each index is a different set of vertical measurement along a single vertical profile and has a time and location of collection. |
| **Tidegauge** | – **Dimensions:** (id_dim, t_dim)<br>– **Coordinates:** time, longitude, latitude | Contains time series data collected at a tide gauge location, e.g. sea level data. This is a child of the *Timeseries* class, which is in turn a child of the *Indexed* class. |
| **Track** | – **Dimensions:** (t_dim)<br>– **Coordinates:** time, longitude, latitude | Contains along-track data such as from an altimeter. Data is ordered along a single track, which lies along the *t_dim* dimension. |
| **Glider** | – **Dimensions:** (id_dim, t_dim)<br>– **Coordinates:** **time, depth, longitude, latitude** | Contains data along a glider path. Similar to Track, however there is an additional depth coordinate and index dimension. |

 ## 1.2  Five Principles for Validation

There are five key principles that any validation workflow should adhere to and integrating COAsT into a workflow can help satisfy them:

1. **Scales with size of data**

    As the power of computing resources increases, model configurations are being developed with higher resolutions and smaller time steps, amplifying the technical challenges associated with analysis and validation. A long-term problem is the storage of data, which must be tackled through careful choices of output and file compression. However, a more immediate issue is how to analyse this data, when there is no hope of being able to fit it all in a processor's memory. For a validation system to be truly flexible, it should be able to scale easily alongside these increasing data sizes. COAsT is able to deal with this problem by building upon Python packages such as Xarray (Hoyer and Hamman, 2017) and Dask
    (Rocklin, 2015), giving the user access to lazy loading[2], data chunking[3] and parallel programming.

   By integrating Dask and Xarray into COAsT, the user has access to a powerful system that provides lazy loading, chunking and parallel code. As COAsT makes use of xarray and Dask, the user also has access to important data analysis tools such as lazy loading (or asynchronous loading), chunking and parallel computation. Lazy loading means that data is not brought to memory until needed, which works well with chunking, where an analysis is performed on smaller
    blocks of data rather than the whole dataset at once. Chunks of the data can be analysed either in series or in parallel by utilising multiple CPUs.

2. **Data source independence**

   Validation code should work quickly and easily with modelled or observed data from various input sources. For example, performing an analysis of output from different numerical models or making a comparison between model data and
    observation datasets from different sources and in different input formats. COAsT allows the user to abstract out the data source by standardizing array structures, dimension names, variable names and basic manipulation routines. Once data has been ingested into COAsT, the source of the data no longer matters and analysis scripts can be applied quickly to any data with compatible structure.

3. **Reproducible and citable**

    Ensuring that validation analyses are easily reproducible means that the same workflow can be applied quickly (or automatically) to an ensemble of model runs or different configurations. This can save time and resources during the phase of model development where physical parameterizations are being chosen and tuned. In addition, this aids in knowledge and code sharing with other researchers who wish to perform an equivalent analysis on their own data. Having reproducible code is not only important for the current state of a code base, but also for all of its history. In other

---

[2]Lazy loading: https://en.wikipedia.org/wiki/Lazy_loading [last accessed 5 Apr 2023]

[3]An overview on Chunks: https://docs.dask.org/en/stable/array-chunks.html [last accessed 5 Apr 2023]

words, it may be important to rerun a historical analysis using the validation workflow, however the code may have since been modified. COAsT uses git for version control alongside a DOI for each release version of the package. This means the exact version of COAsT used for a validation script can be tracked and re-used if necessary. Broader application of the reproducibility principle as applied to regional ocean modelling is discussed in more detail in Polton et al. (2023).

4. **Expandable codebase**

No analysis is static and will change over time, with modifications to existing analyses and the addition of new metrics and methods. Indeed the validation presented in this paper is only a foundation and has many avenues for expansion. It is therefore vital that code is written in such as way as to make changes and expansions by a community of contributors as easy as possible: COAsT uses a public and open source license; well written documentation and user tutorials; and the documentation contains guidelines for programming practices for contributors. Having a mechanism for unit testing old and new contributions (with good coverage) is fundamental to maintaining a working codebase. A system of testing is an essential tool for rapid error trapping, when creating a robust codebase by multiple authors, against a backdrop of evolving module dependencies. For COAsT, unit tests cover approximately 67% of the package at the time of writing. Suitable coding structures can also assist, for example COAsT makes use of object orientated coding structures, which facilitates independent contributions.

5. **Objective scoring**

Deriving metrics which minimize room for interpretation is essential when comparing large numbers of model configurations. Where possible, a validation should not be done subjectively, for example visual comparisons of time series. Instead, well defined scoring metrics should be used. This might include commonly used integrated metrics such as the Mean Absolute Error (MAE), Root Mean Square Error (RMSE) and correlation or more complicated probabilistic metrics such as the Continuous Ranked Probability Score (CRPS) (Matheson and Winkler, 1976). Some types of analyses also require different approaches such as extreme value analyses (e.g event based hazard validation).

## 2 Workflow Demonstration: Data

### 2.1 Model data

We showcase our validation workflow using two different model configurations. Both are built using the NEMO model framework (Nucleus for European Modelling of the Ocean) but at versions 3.6 and 4.0.4 respectively (Madec and Team, 2016, 2019). They belong to the Coastal Ocean (CO) model series, which are used operationally by the UK Met Office for the Northwest European Shelf. At the time of writing, CO7 is the most up to date published version of the configuration (Graham et al., 2017), with CO9 being under development[4]. In this paper, we provide a validation and intercomparison of the CO7 configuration and what we will call CO9p0 configuration. CO9p0 is the first configuration iteration during the development of what will be called

---

[4]CO8 was skipped for reasons of consistency.

**Table 3.** Summary of Model configuration and forcing data for CO7 and CO9p0

| | |
|---|---|
| **Bathymetry** | EMODnet, September 2015 release |
| **Lateral Boundary Conditions** | 3D $T$ and $S$, barotropic velocities and external SSH + tidal forcing. |
| **Atmospheric Forcing** | ERA-Interim (Dee et al., 2011) |
| **Atlantic Boundary data** | $1/4^\circ$ Global Seasonal Forecast System (GLOSEA) version 5 (MacLachlan et al., 2015) |
| **Baltic Boundary data at $12^\circ$ E** | From $1/60^\circ$ GETM model of North Sea and Baltic (Gräwe et al., 2015) |
| **Tidal Forcing** | TPX07.2 (Egbert and Erofeeva, 2002) |
| **Grid** | AMM15 (1.5km) C-grid with rotated pole as described in Graham et al. (2017) |
| **Initial Condition CO7** | Initialised at rest on January 1st 1985, using $T$ and $S$ from ORCA025 configuration (Megann et al., 2014). |
| **Initial Condition CO9p0** | Started from CO7 restart file from January 1990. Analysis period starts 2004 |
| **Vertical Coordinates** | 51 vanishing quasi sigma levels as in Siddorn and Furner (2013) |
| **Riverine Forcing** | Daily climatology of gauge data averaged for 1980–2014 UK data |
| **Output frequency** | Daily mean 3D temperature and salinity fields. Hourly 2D sea surface height |

CO9. It is intended to perform very similarly to CO7; capturing the change from NEMO v3.6 to NEMO v4.0 with as few other changes as possible. However differences do arise from structural changes introduced into the model. Known differences include: 1) Bulk forcing implementation: We used the "NCAR" algorithm in CO9p0 in an attempt to closely match the CORE bulk forcing algorithm in CO7 (Large and Yeager, 2009); 2) Lateral diffusion of tracers: We attempt to replicate the constant value used in CO7; 3) Lateral diffusion of momentum: This *is different* between model runs. For stability purposes we deviate

from a constant value (as used in CO7) and use the NEMO4 option that varies diffusion according to grid scaling and local velocity; 4) Tracer advection: In CO7 the Total Variance Dissipation (TVD) was used. In NEMO4 the closest equivalent is the Flux Corrected Transport scheme (FCT), which is set to 2nd order in horizontal and vertical directions. 5) Initial conditions differ. The grid and external forcings, as summarised in Table-3 are the same.

## 2.2    Observation Data

The validation workflow presented below uses in-situ observations of sea surface height, temperature and salinity. For validation of tides and non-tidal sea surface height, we use time series and harmonic analyses from tide gauges around the model domain. For validation of tides specifically, tidal amplitudes and phases are derived from time series taken from multiple sources, and some of the raw timeseries data are no longer available. These eclectic sources include tide gauges, moorings and bottom pressure sensors with a very large range in analysis length. Using such a variable dataset allows us to test the harmonic

matching and uncertainty discussed in Section-3. To validate high-frequency sea surface height time series, we use quality controlled tide gauge data from the GESLA database (Woodworth et al., 2016). The locations for these quality-controlled locations are shown in Figure-1. For validation of temperature and salinity, vertical profiles from the EN4 database (Good et al., 2013) are used.

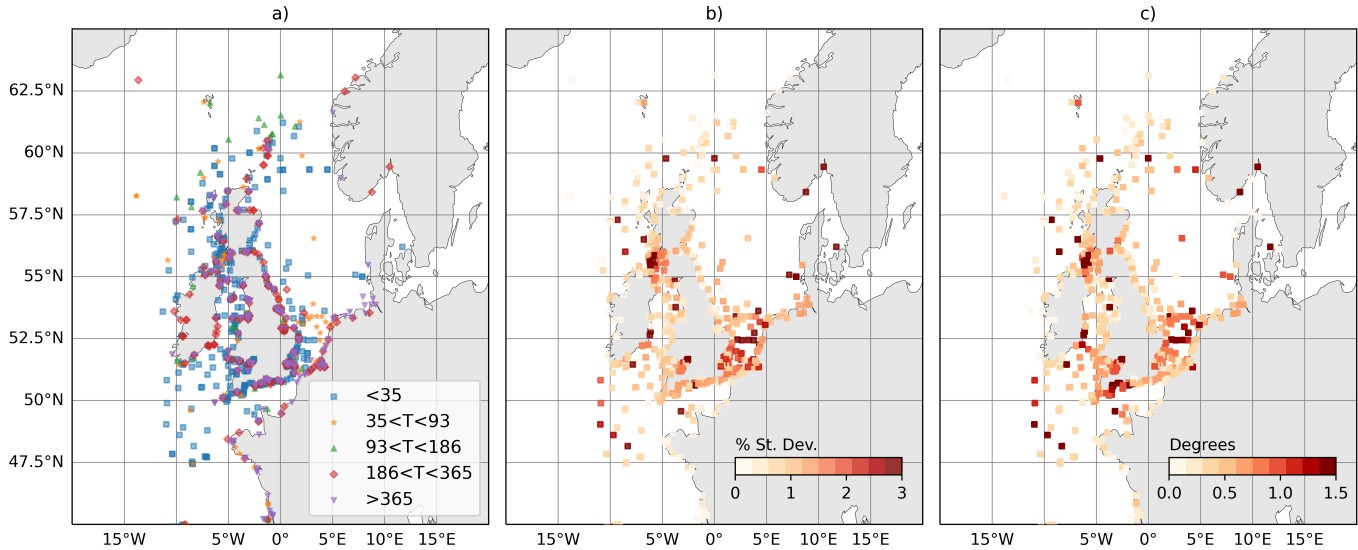

**Figure 1.** Estimation of uncertainty in M2 amplitude and phase resulting from different analysis lengths and constituent sets. **a)** Analysis lengths (days) used for each observation location. **b)** Ensemble standard deviation in M2 amplitude as a proportion of amplitude (%). **c)** Ensemble standard deviation in M2 phase (degrees).

## 3  Workflow Demonstration: Validation against tide gauge data

The high frequency measurements made by tide gauges are useful for validating modelled sea level processes such as tides and non-tidal residuals. In shorter term model simulations, simulating these phenomena accurately can have significance for forecasting (e.g. coastal sea levels) and oceanographic processes (e.g. internal waves, enhanced mixing). On longer time scales the data can be filtered to obtain longer period signals such as trends due to sea level rise. In this section, we use the *Tidegauge* class within the COAsT framework to demonstrate validation of tides and non-tidal residuals. We can use this class to represent

and quickly manipulate time series data across multiple locations, which lie along the same time coordinate. COAsT also contains provision for extracting the nearest time series from 2-dimensional model SSH data for each tide gauge location. Using this method provides two instances of the *Tidegauge* class: one for the observed time series and one for the extracted modelled time series. These can be easily compared and analysed.

For high frequency data, the estimation of the tides is a vital step for validation of sea surface height (SSH) in our regional

models. In both the observations and the model, these periodic oscillations can both be validated and removed to allow for an analysis of residual signals (non-tidal residuals). Tidal signals exist across a large spectrum of frequencies. Though the most dominant periods are diurnal or semi-diurnal, many shorter or longer periods also exist. These signals of differing periods are known as tidal constituents and are the result of different tidal forcing and periodic interactions. Non-tidal residual signals can be generated by many processes but in coastal regions the most significant are generated by atmospheric processes such as

atmospheric pressure gradients, wind generated currents and waves (Vogel, 2015).

To validate and remove the tides, we use a harmonic analysis to estimate tidal amplitudes and phases. This method aims to decompose the sea level signal into a sum of sinusoidal harmonic constituents, with differing frequencies, amplitudes and phases (Vogel, 2015). By determining constituents with predefined frequencies, amplitudes and phases can be estimated using a harmonic analysis of SSH time series. A least squares optimisation is commonly used to do this (see also Fourier methods), where a set of amplitudes and phases are obtained which together form a best fit to the data. This methods also allows for each constituent to be multiplied by a time-varying nodal correction, which is important for extrapolating tidal predictions.

## 3.1 Harmonic Analysis

### 3.1.1 Matched Harmonic Analysis (MHA)

When both validating the tidal signal and removing it from the total signal, it is important that the harmonic analysis of modelled and observed data is equivalent. Each set of harmonics may be calculated from time series of different lengths, time periods and using different constituent sets. Furthermore, differences in software methods can add more variance in the output, creating some uncertainty in any comparisons between the model and observations. These variations can cause differences in the amplitudes and phases of estimated harmonics, leading to difficulties comparing harmonic information from model and observations. Similarly, these harmonic differences may propogate into timeseries obtained through the subtraction of tides (e.g. non-tidal residuals for storm surge validation).

Below, we outline the steps used in our validation workflow to ensure that analyses are equivalent. When these are true, we call this a Matched Harmonic Analysis (MHA).

1. Obtain observed time series and extract model SSH time series at the nearest model grid cell.

2. Subsample or interpolate both time series to be on the same frequency – for example hourly.

3. Where there are missing or flagged data in the observations, also remove these data from the model time series.

4. Ensure both time series start and end at the same time.

5. Perform a harmonic analysis of each time series, using the same constituent set. The constituent set should be determined using the Rayleigh Criterion (Vogel, 2015).

6. For each time series, reconstruct a tidal signal from the estimated tidal harmonics. Subtract these signals from each time series to obtain non-tidal residuals.

By not following these steps, significant uncertainty may be introduced into any comparisons between modelled and observed harmonics. This uncertainty will also be propagated into comparisons of non-tidal residuals. At the time of writing, COAsT contains a number of routines which make applying the MHA to modelled and observed data quick and easy. By representing both observed and modelled timeseries in an instance of the *Tidegauge* class, we can quickly slice out equivalent time periods, interpolate to the same frequency and match missing values across each dataset.

### 3.1.2   Harmonic uncertainty estimation where MHA is not possible

In some cases, it may not be possible to follow all of the steps above, e.g. when time series are short or time periods do not coincide. For these cases, we instead attempt to quantify and apply this uncertainty. This is true for the large dataset of harmonics we use for tidal validation (described in Section-2.2). To estimate uncertainty in observed harmonics, we use modelled SSH time series on an hourly frequency. We have extracted these time series at the nearest model points to each location in the observed dataset described in Section-2.2. For each observation location the analysis lengths have been identified from observations and these are shown in Figure-1(a). For each model SSH time series, an ensemble of harmonic analyses is performed. Each member of the ensemble contains a harmonic analysis with a length defined by the analysis length of the corresponding observation and an appropriate constituent set according to the Rayleigh Criterion. The ensemble standard deviation is then calculated at each location for each constituent and used to define the observation uncertainty. From an initial set of analyses, analysis length and the time period of analysis were found to be two of the most influential generators of uncertainty. When isolated, the chosen constituent set was also found to have an effect, although not as large.

Results from this analysis are shown for M2 in Figure 1(b)-(c). For amplitude, they are expressed as a percentage of the amplitude obtained from a 10-year harmonic analysis. The standard deviations in amplitude can reach over $3\%$ at some ports, which can be large at locations where M2 is also large (e.g. the Severn Estuary). Differences in phase appear less consequential, reaching a few degrees. This analysis shows that there is a significant degree of variability in harmonic analyses when time series of differing length and period are used. In turn, this variability can become uncertainty when comparing harmonics from model and observed data. Using an MHA can reduce this uncertainty, but where this is not possible the uncertainty must be considered instead. In the next section we show an analysis using both the MHA approach and an application of the harmonic variability estimated here.

### 3.2   Validation of tides

As discussed above, we validate tides using a comparison of tidal amplitudes and phases obtained using a harmonic analysis. The harmonic analysis is obtained using hourly SSH data for the full duration of the model run. SSH time series are extracted from the model at the nearest wet grid point for this purpose. Observations used are described in Section-2.2.

Maps of errors in amplitude and phase for six of the largest semidiurnal and diurnal constituents are shown in Figures 2 and 3. The raw errors are shown in the left two columns. The rightmost columns show the difference between absolute errors for each of the model configurations. As discussed in Section-3.1, we would ideally apply a matched harmonic analysis between the model and observations. However, some historical data are only available as pre-computed harmonics, or the observations fall outside of the simulation window. In these circumstances we can compute an estimated harmonic uncertainty for any comparison between modelled and observationally derived harmonics (as discussed in Section-3.1.2). Here we perform a normal harmonic analysis at each point using all available model data. Then, where differences between the model and the observations are smaller than the uncertainty, they are deemed insignificant and coloured grey in the figures. This is done

independently for each harmonic constituent. By doing this, we mask out points where the model is imperceptibly close to the observations and, most importantly, make no judgements or comparisons on the two models performance at these points.

In the figures, we see that across all constituents and for both amplitude and phase, the spatial distribution of errors between the two models is structurally similar. Visually, the two models look very close, and this is reinforced somewhat by the improvement panels. The errors at many points, especially those away from the coast, are similar enough that they lie within the observational uncertainty. Where there are significant differences, they are small – being on the order of a few centimeters or degrees. The models are closest for K1 amplitude and phase, where there are very few significant differences. Notably, P1

shows more significant differences than the other constituents, especially for phase.

### 3.3    Validation for non-tidal residuals

Whereas harmonics are useful for validating periodic signals due to tides, they do not capture other motions, such as those caused by interactions with the atmosphere. Storm surges are the most impactful of these in coastal regions, and can be measured with metrics such as the non-tidal residual and skew surge. Therefore, validation of the full SSH signal, and its

components in the time domain, is important. In this section we consider the full SSH signal as well as a validation of non-tidal residuals. Here, we again use the functionality within the *Gridded* and *Tidegauge* classes of the COAsT package. In particular, we are able to use these classes to extract the nearest time series from the *Gridded* object, subtract tides from the signal, calculate some simple metrics and perform a basic extreme value analysis.

     As discussed above, to derive non-tidal residuals (NTR) a tidal signal must be subtracted from the full SSH signal. In

operational scenarios, a separate tide-only run may be performed to get a time series of tides at all model locations, which can be used to estimate non-tidal residuals. However, interactions between the tidal and non-tidal components (for example see Prandle and Wolf (1978)) mean that this may not be an accurate way of removing the tide and is likely to leave a significant periodic signal in the resulting residual. Therefore, it is recommended that a harmonic analysis of the full signal is performed and the reconstructed water levels subtracted. To avoid uncertainty in tidal harmonics propagating into the residual signal, we

only use data for which the matched harmonic analysis (described in Section-3.1) can be applied.

     Once non-tidal residuals have been derived, various statistics can be estimated. Errors and absolute errors (or MAE/RMSE) can be sensitive to residual tidal constituents that may not have been completely removed by the harmonic analysis process. This is especially true when non-tidal residuals are small such as during calm atmospheric conditions like a high pressure system. In addition, it is often the largest non-tidal residuals which we are most interested in modelling accurately, as they have

the largest impact. The COAsT package currently has a number of options available for extreme value analysis of non-tidal residuals. As a part of the validation workflow, the package will calculate:

1. Daily and Monthly maxima at all tide gauge locations. Count the number of each over specified thresholds

2. Identify independent peaks in the signal and count those over specified thresholds.

3. Integrate the total time spent by a signal over specified thresholds.

This is a simple extreme value analysis and can be expanded upon in future. For example, the application of a generalized extreme value distribution to signal peaks or the generalized pareto distribution to daily or monthly maxima.

     Figure-4 shows the error in the standard deviation of total water level for each of the two models along with the difference in errors between the two models. Where positive, this difference indicates that CO9p0 performed better (i.e. the absolute standard deviation errors were smaller). This metric can be used as a proxy for the atmospherically influenced tidal range tidal

range, averaged over time. It is a good way to measure the error across all harmonic constituents. Both models have similar spatially, overestimating standard deviations at most locations. The most notable difference between the two models is in the Severn Estuary, where tidal ranges are large. Here, the CO9p0 configuration performs much better. This is likely because of the parallel improvements seen in the amplitude of the largest constituents (see Figure-2).

     Figure-5 shows the correlations between modelled and observed NTR time series over the full duration of the run. This

gives us information about how well the modelled and observed signals move together, as well as the timing of the non-tidal residuals. In this case, the CO9p0 model performs better at the majority of locations, especially those in estuarine areas such as the Severn and Thames estuaries.

     Figure-6 shows two examples of extreme value statistics for the non-tidal residuals, expressed as a function of thresholds. Here, non-tidal residual thresholds have been defined between $0\,\mathrm{m}$ and $1.5\,\mathrm{m}$. Figure-6(a) shows the number of modelled

independent peaks over each threshold divided by the number of observed peaks. For example, a value of 0.5 indicates that the model generated only a half of the number of peaks over a given threshold as were observed. Peaks were defined as independent maxima, separated by at least 12 hours, to avoid double counting the same events. Figure-6(b) shows the total time spent over each threshold, again as a proportion of that in the observations. Together, these two figures show us how well the models are able to capture large events such as storm surges over the period of the model run. However the figures also

show that both configurations are underestimating large events, and the larger the observed event gets, the more their number is underestimated. It also seems that neither model is sufficiently sensitive to atmospheric forcing and that coastal effects, such as resonance, are not being adequately represented.

## 4   Workflow Demonstration: Validation against profile data

     To validate temperature and salinity in the model configurations, we compare them to in-situ profiles from the EN4 database

(Good et al., 2013). As discussed, the *Profile* object within the COAsT framework is well suited to handle this type of data. The routines within it allow us to read from common profile databases such as EN4 and World Ocean Database, represent multiple profiles in a single standardised object, manipulate this data in time and space and extract comparable model profiles. Similarly to the analysis of tide gauge data above, the extracted model data may also be represented using a *Profile* object for quick and easy comparisons.

Model profiles are extracted using multiple independent interpolations, both in space and time. The horizontal and vertical interpolations are also treated independently. Horizontal interpolations can be done quickly using either a nearest neighbour or bilinear approach. For the analyses below, we have used a nearest neighbour interpolation, taking the nearest model wet

point to each observation. However, in some cases this may result in a model cell being chosen that is too far from the observation location, such as near complex coastlines. These points are few, but to avoid them having an effect on the analysis, all interpolated points further than $5\,\mathrm{km}$ from from the nearest observation location are omitted. This horizontal interpolation results in a time series of data at every model depth. This is then interpolated onto the measurement time of the nearest profile using a linear interpolation.

At this point we have a model profile for each observed profile, however the data still lies on model depth levels so a vertical interpolation is still required. Since this is for comparison purposes, it is useful to interpolate both the model and observations onto the same reference depths at each location. This can be done in a single interpolation step, i.e. interpolating directly onto a set of reference depths, however this can cause problems where the vertical coordinate density varies drastically between the two. Errors in the interpolation may be falsely interpreted as errors in the model. This is likely to be most prominent in areas with high vertical gradients in variables like density, for example near a pycnocline. We can reduce this problem by doing the vertical interpolation in two steps:

1. Interpolate the model onto observation depths if the latter are sparser, or vice versa.

2. Interpolate both the observations and interpolated model data onto the same reference depths.

The COAsT package is able to interpolate sets of profiles onto reference depths, or onto the depths of another *Profile* object. This allows for flexibility with vertical interpolation and applying the two step process discussed above.

Now that we have a set of modelled and observed profiles on the same reference depths, we can do a comparison. At higher resolutions, where smaller features may be resolved, traditional metrics such as the Mean Absolute Error and Root Mean Squared Error may not be the best options for validation. These metrics can result in a double counting effect for small scale features such as eddies. For example, a higher resolution model may resolve a certain scale of eddy when a lower resolution model may not. However, if an eddy is not at the right place at the right time, then MAE will count both the error at the observed eddy location and at the modelled eddy location. To counter this, a probabilistic metric should be used instead. To do this, we have used the Continuous Ranked Probability Score (CRPS) (Matheson and Winkler, 1976) at the ocean surface. This is generally used with an ensemble of model states to compare the Cumulative Distribution Functions (across ensembles) of the model and a single observation at a specific location. However, when a small enough neighbourhood around a location is taken, it can also be considered sufficiently random and used in place of the ensemble. This is the formulation used here and we apply it to both surface temperature and salinity:

$$CRPS(F,x) = \int_{-\infty}^{\infty} (F(y) - \mathbb{1}(y-x))^2 dy, \tag{1}$$

where $F$ is a cumulative distribution function (CDF), $x$ is a single observation and $\mathbb{1}$ is a Heaviside function that is equal to 1 where its argument is greater than or equal to zero, and 0 otherwise. The CDF $F$ is derived from all model values within some predefined radius around the observation $x$, as described above. $F(y)$ is then the value of this CDF for a single element

of the model radial dataset. The difference is integrated spatially over data from the radial neighbourhood. More intuitively, the CRPS can be thought of as the mean square error between modelled and observed CDFs.

In order to identify regional differences in model errors, we average estimated errors into regional and depth bins. The regions used in this paper are shown in Figure-7. The validation workflow within COAsT includes flexible routines within the *mask_maker* class for averaging errors both into regional areas and into depth bins. This class contains predefined region definitions that help standardize spatially aggregated metrics. This is a useful method for obtaining averaged objective error metrics whilst still targeting specific areas of the domain. Figures $8-9$ show the mean absolute errors in depth levels for levels less than $150\,\text{m}$. The absolute error metric allows us to see how each model performs, for temperature and salinity, at different depths and in each region. In addition, these plots are split into summer (JJA) and winter (DJF) seasons. We can see that both of the model configurations studied in this paper are similar at all depths and most regions. Figures 10-11 show CRPS values for SST and SSS respectively. In this case, we see similar CRPS values across all radii. For SST, the CO7 configuration performs best (i.e. smallest values) for most regions. The opposite is true for SSS. For both variables, the Norwegian Trench and Kattegat generally have the highest (worst) CRPS values.

## 5  Further Discussion

In this paper, we have presented the underlying principles used to develop a new standardised workflow for validating model data. The principles introduced aim to ensure that assessments are scaleable with data size, independent of data source, reproducible, have a codebase that is easily expandable and use metrics which provide objective comparisons. They aim to ensure that any analysis of model data can be easily applied, cited, reproduced and interpreted. Alongside our discussion, we showcased an existing workflow for validating modelled sea surface height, temperature and salinity against tide gauge and profile observations. In the examples shown, we have adhered to the principles set out in this paper. We used the COAsT Python package, which offers a set of standardised oceanographic data structures which are ideal for comparing model data with observation. The package, along with its components (especially Xarray), are important tools, allowing us to adhere to the assessment principles with ease.

This kind of validation framework can be used as an integral part of the model development process. Such a process is often iterative, with the tuning of various processes and parameters being necessary. This kind of workflow, again with the added benefits of the standardised COAsT framework, can be run quickly and with minimal changes for each new set of model output. In addition, it is recommended that model configurations are evaluated and tested using the same validation and the same sets of metrics for consistency. The version control and DOI system used by COAsT also means that any validation code may be reverted back to in the future if necessary. For example, this may be needed if there are differences or errors in output that need to be evaluated.

The assessment metrics presented in this paper are by no means exhaustive, and serve as a demonstration and foundation for future additions. As discussed throughout this paper, a real advantage of our approach is that new assessments may be added to the analysis in an easy, modular fashion, whilst preserving the reproducibility of past iterations of the code. New validations

may be designed to incorporate more sources of observations, new metrics and diagnostics, new variables and comparisons to other models.

The philosophy introduced in this paper has been used for our own development with great success. We present these reflections in hope that they are useful to the wider community.

**Author Contributions**

DB wrote the primary manuscript draft with additional contributions from JP, EOD and JW. Experiment design was lead by DB, with additional contributions from JP, EOD and JW. Development of model configurations was lead by EOD, with contributions from DB and JP.

*Acknowledgements.* This work was conducted through the Weather and Climate Science for Service Partnership (WCSSP) India, a collaborative initiative between the Met Office, supported by the UK Government's Newton Fund, and the Indian Ministry of Earth Sciences (MoES).

**Appendix:  Code Availability Statement**

The scripts used in this paper are made openly available as a GitHub respository: https://github.com/JMMP-Group/NEMO_validation. The COAsT toolbox is also available from GitHub: https://github.com/British-Oceanographic-Data-Centre/COAsT. At the time of writing, the versions used are available as Zenodo DOIs: COAsT (https://doi.org/10.5281/zenodo.7799863) and NEMO_validation (https://doi.org/10.5281/zenodo.7949115).

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

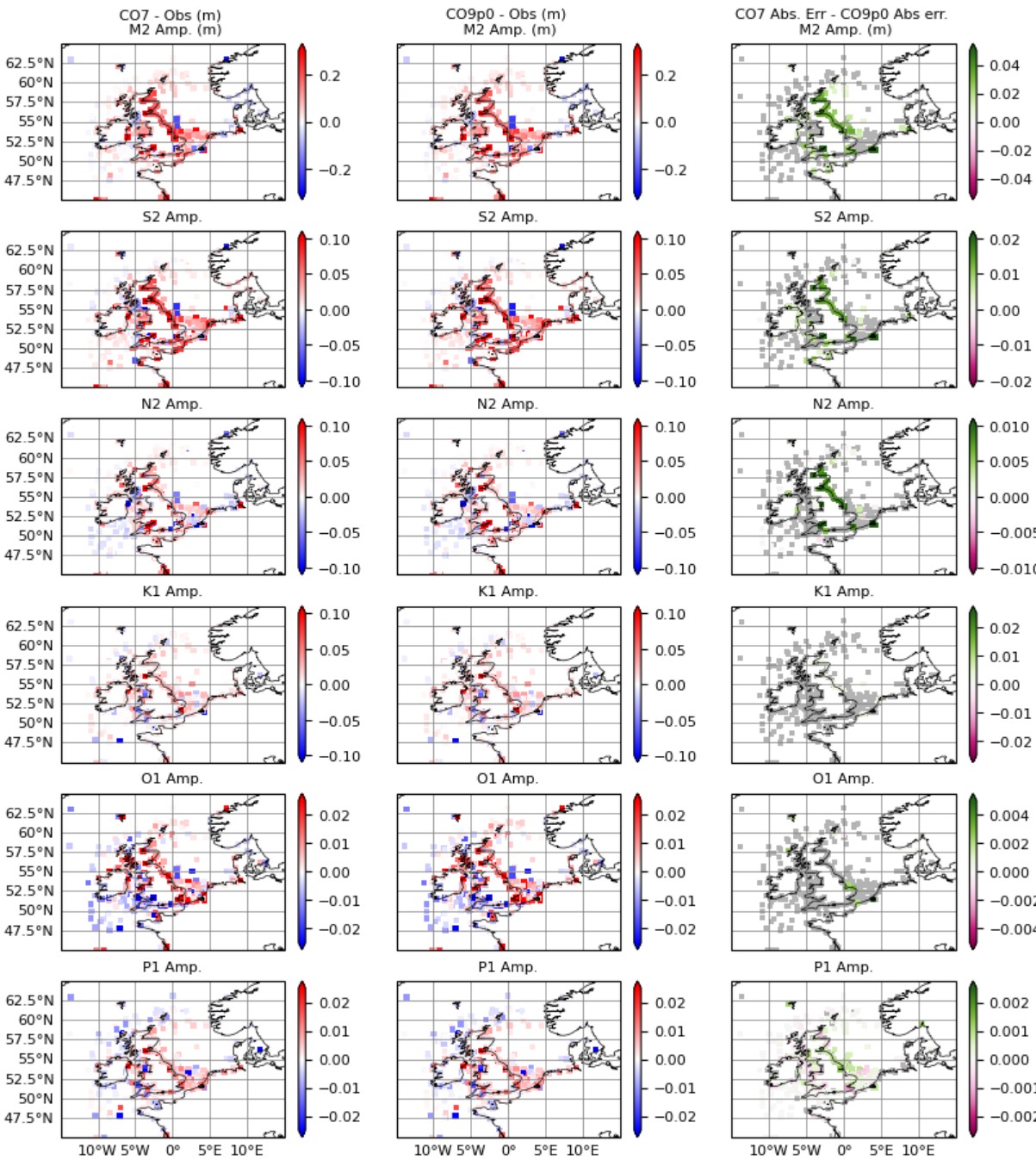

**Figure 2.** Errors in amplitude (m) for **left:** the CO7 model run, **middle:** the CO9p0 model run. Errors are calculated as model minus observations, meaning positive values indicate overestimation by the model. The rightmost column shows the difference in absolute error between the two model runs, calculated as CO7 - CO9p0. Constituents are obtained using a normal harmonic analysis and then differences are masked according to harmonic uncertainties. Grey values show locations where differences were smaller than the estimated uncertainty in the model-observation comparison.

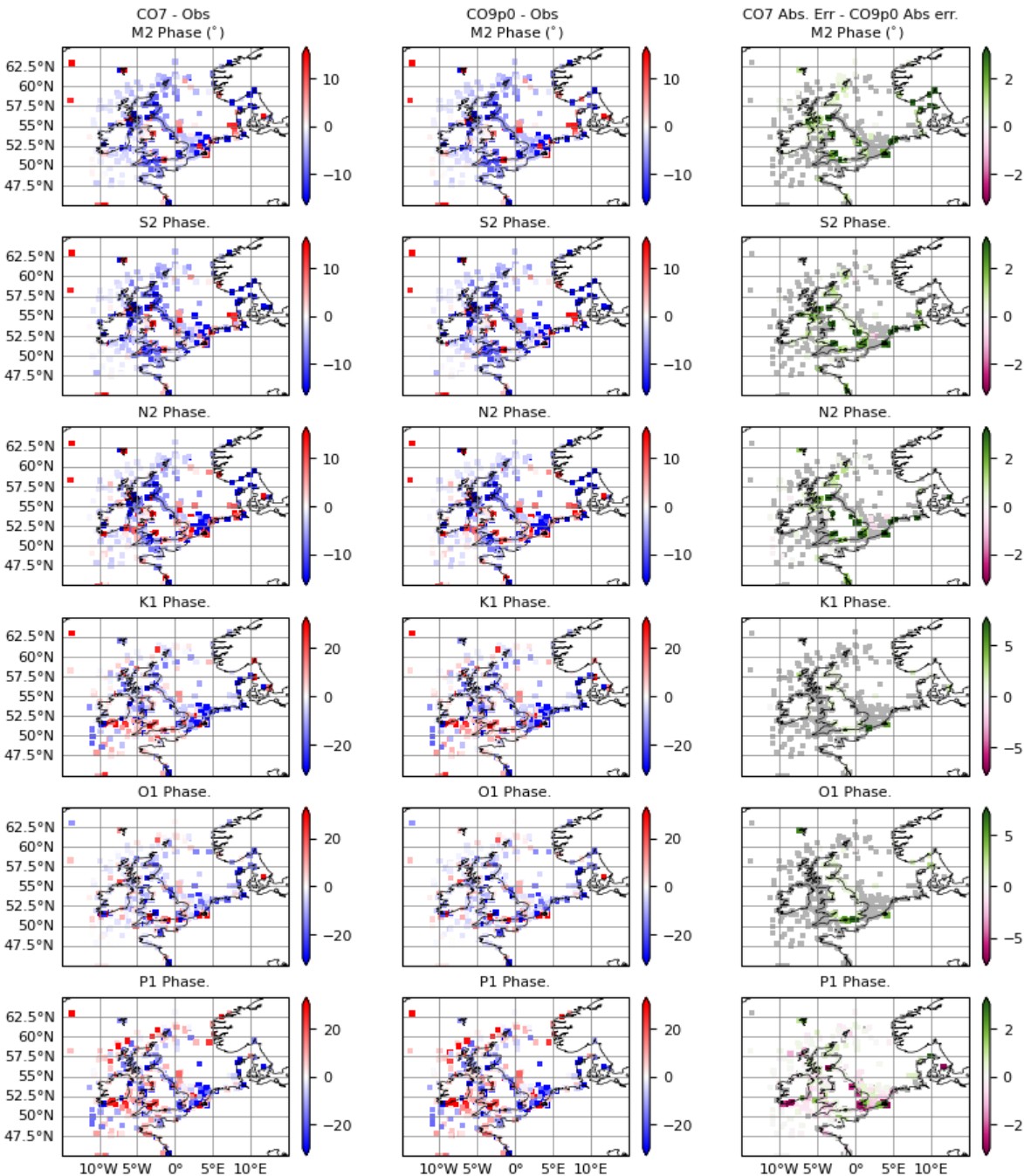

**Figure 3.** Errors in phase (degrees) for **left:** CO7 model run, **middle:** the CO9p0 model run. Errors are calculated as model minus observations, meaning positive values indicate overestimation by the model. The rightmost column shows the difference in absolute error between the two model runs, calculated as CO7 - CO9p0. Constituents are obtained using a normal harmonic analysis and then differences are masked according to harmonic uncertainties. Grey values show locations where differences were smaller than the estimated uncertainty in the model-observation comparison.

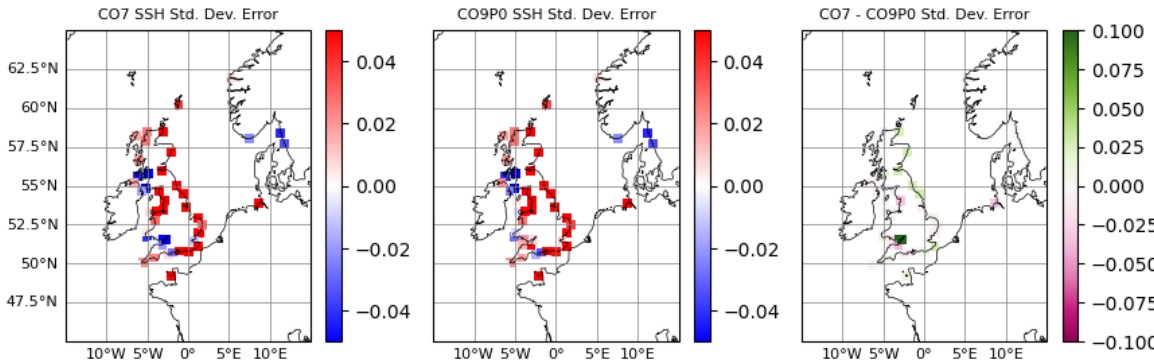

**Figure 4.** Errors in Total Water Level standard deviations, calculated over the 10 year model run for the CO7 and CO9p0 configurations. The two panels on the left show the errors for the CO7 and CO9p0 configurations.

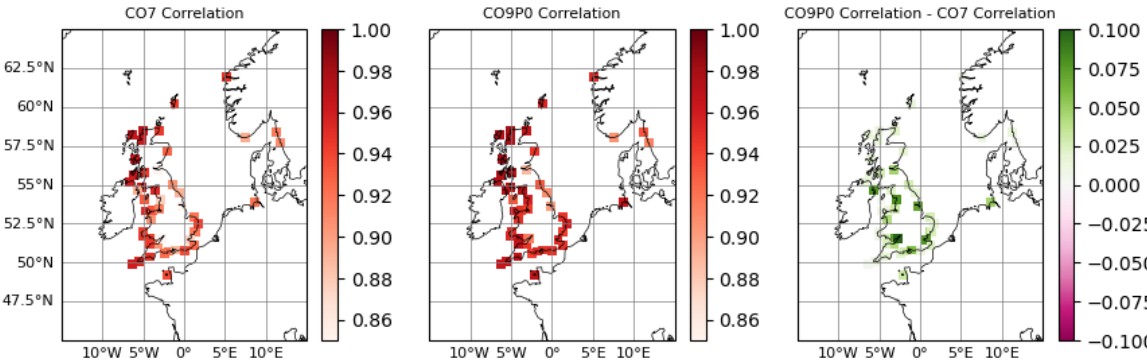

**Figure 5.** Correlations between modelled and observed non-tidal residuals at tide gauge locations. The left and middle panel show correlations for the CO7 and CO9p0 model runs respectively. The right panel shows (where positive) where the CO9p0 run had higher correlations than the CO7 run.

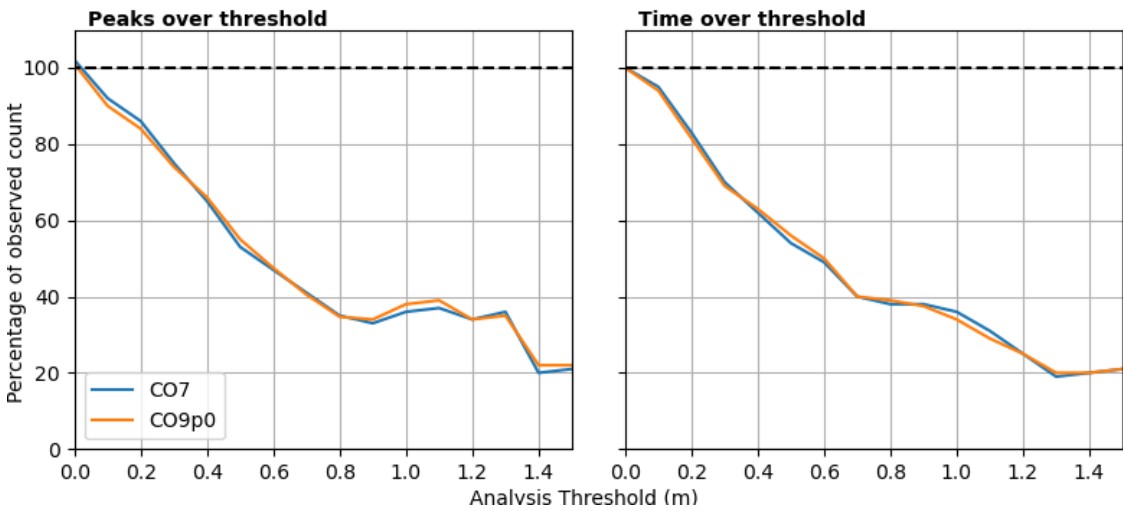

**Figure 6.** Threshold statistics for non-tidal residuals in the CO7 and CO9p0 model configurations. **Left:** The number of independent peaks over a given threshold, as a proportion of the number of peaks in the observations. **Right:** Total time spent over threshold, as a proportion of the time spent by observations. Non-tidal residuals are derived by subtracting a tidal time series calculated form a Matched Harmonic Analysis.

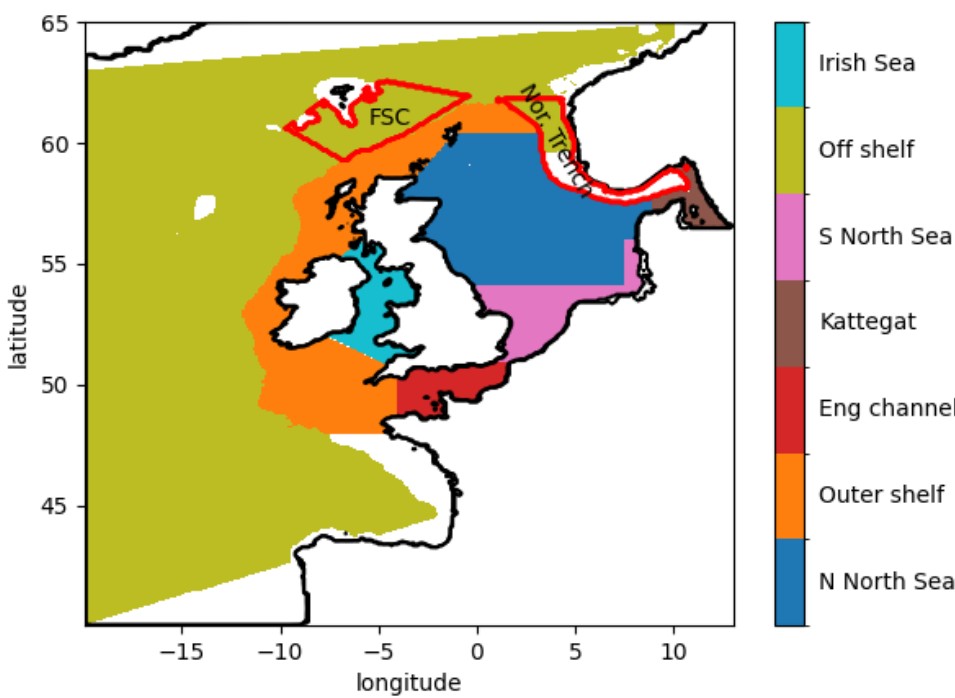

**Figure 7.** Illustration of the different averaging regions used in this study.

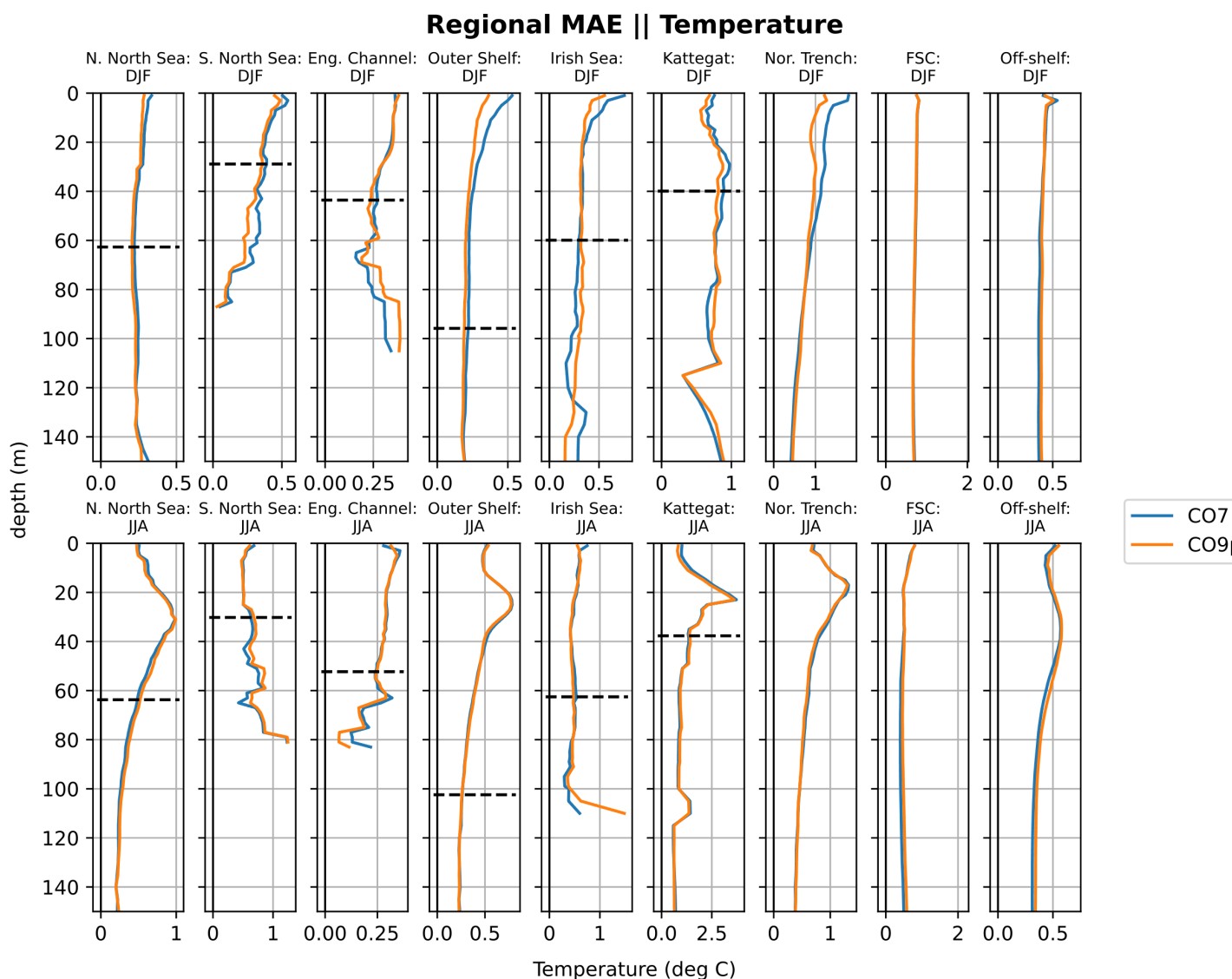

**Figure 8.** Absolute temperature errors with depth for regions in the AMM domain. Comparisons made between the CO9p0 and CO7 model runs. The black dashed horizontal lines show the mean bathymetric depth across the profiles used in each region (where displayed profile is deep enough).

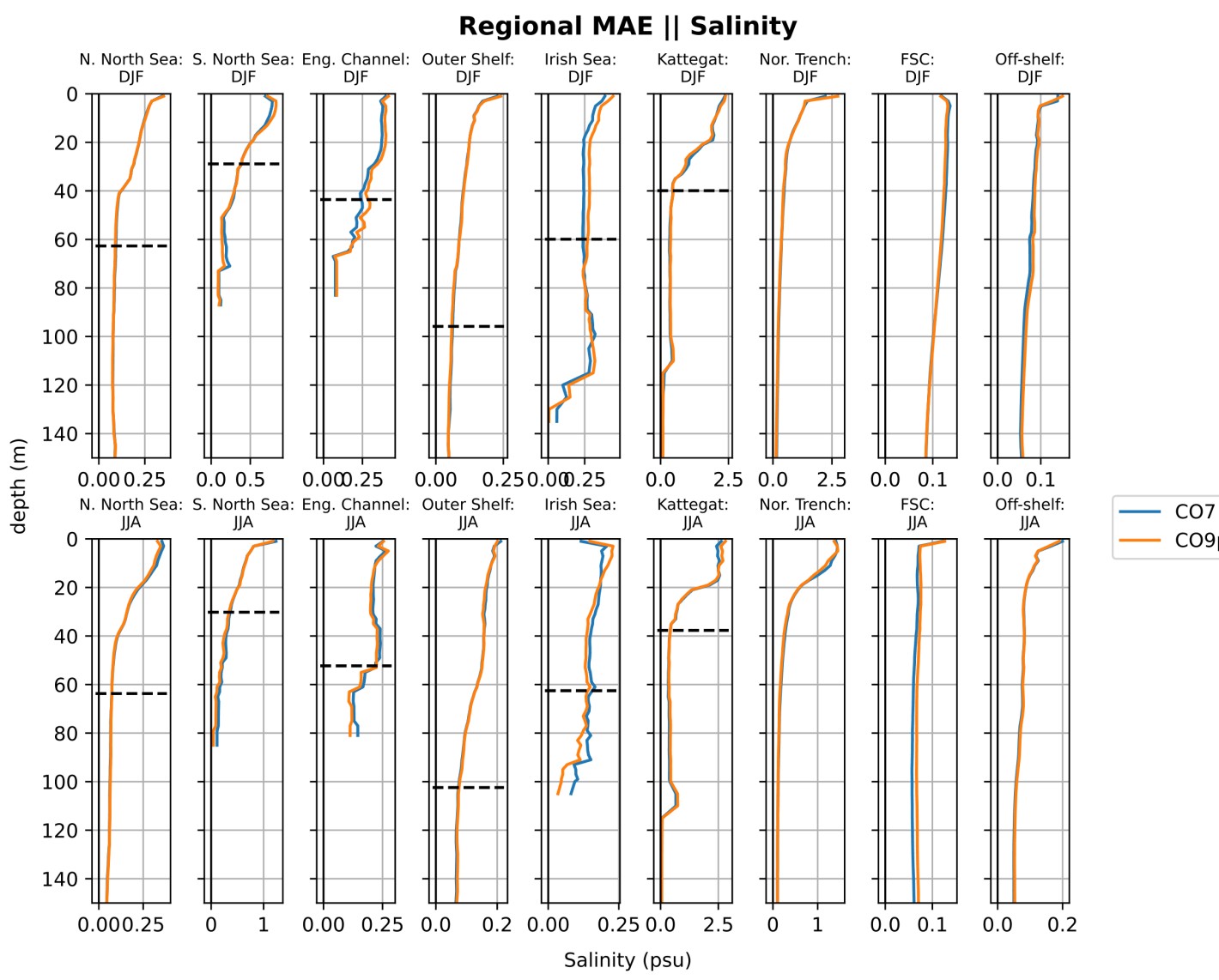

**Figure 9.** Absolute salinity errors with depth for regions in the AMM domain. Comparisons made between the CO9p0 and CO7 model runs. The black horizontal lines shows the mean bathymetric depth across the profiles used in each region (where displayed profile is deep enough).

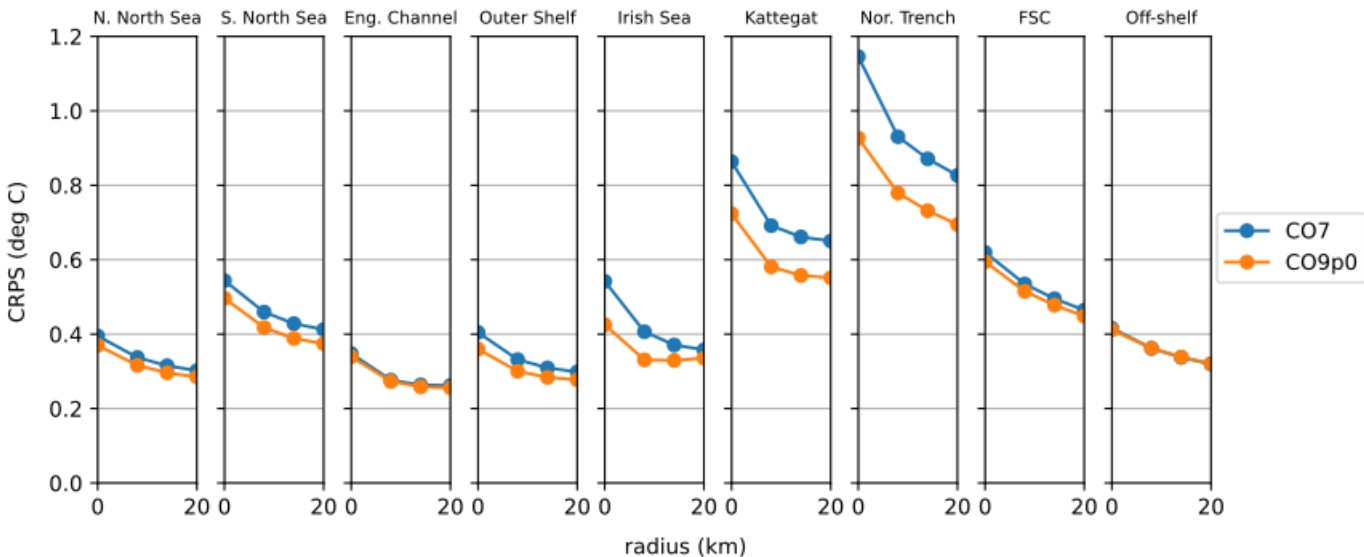

**Figure 10.** SST Continuous Ranked Probability Score (CRPS) for eight regions in the AMM domain. Comparisons made between the CO9p0 and CO7 model runs.

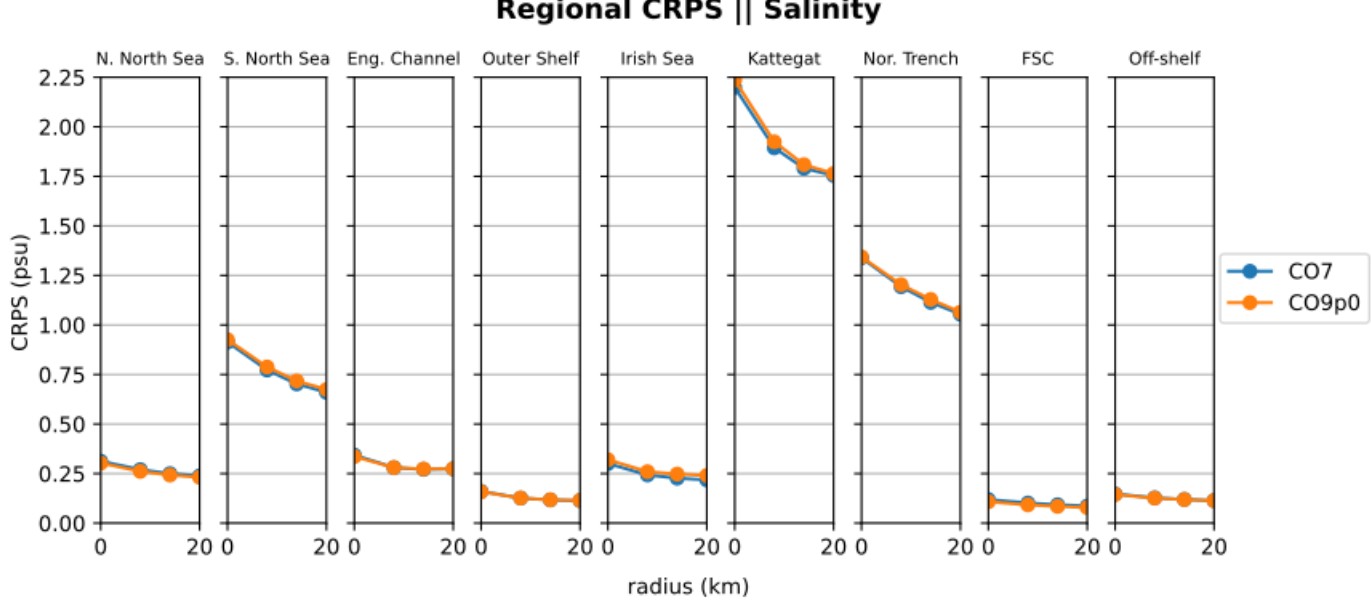

**Figure 11.** SSS Continuous Ranked Probability Score (CRPS) for eight regions in the AMM domain. Comparisons made between the CO9p0 and CO7 model runs.