# Peer review of "Using the COAsT Python Package to Develop a Standardised Validation Workflow for Ocean Physics Models"

_Geoscientific Model Development, 2022_

## Author Comment (AC1)

We'd like to thank both reviewers for their comments, which have been very useful and encouraging. We have responded to them all individually below. However here, we have provided a quick summary of the larger changes made to the manuscript:

1. Generally, the first half of the paper has been reframed slightly in favour of COAsT. In this paper, we aim to introduce and describe the validation relevant parts of the COAsT package, especially the philosophies which can be integrated into validation workflows.
2. The overview of COAsT now comes earlier in the paper, follows by the validation principles. This latter section then references back to COAsT where relevant.
3. The benefits of COAsT structures have been explained more carefully throughout.
4. Some figures have been updated throughout the paper.

In this document, reviewer comments are in black and author responses are in green.

**REVIEWER 1**

One of my struggles in reviewing this paper has been that it is not clear if it is a documentation paper for the COAsT python package. I do not wish to review the whole of the COAsT package because its goals are a bit nebulous. The scope of this paper is narrower and makes more sense to me, but I still feel that the manuscript could more clearly state that the COAsT python package contains other tools that are not useful for comparing models with observations in this way.

We have reworded and restructured the Introduction section of the paper to hopefully make this clearer. See above.

The list in the introduction to the paper reads like a list of generic values that are important for all scientific code. I hope that it can be better tuned to make it clear why particular choices were taken. The software framework described here puts the data in a very specific format: this specific format is an advantage in this context, and the goal of this work is not to write general code for comparing any model with any observations.

That's correct. The goal is to set out abstracted data classes to be used within validation scripts. The list may seen generic, however the principles are important to model validation workflows. Listing these principles also gives us the opportunity to outline why COAsT is useful for satisfying them.

Major comments:

1. On first read, it was not clear to me that using classes like the "Gridded" class had real benefits. It seemed to me that this data could simply be stored as an xarray dataset, and the relevant dimension names could be input into any plotting or calculation functions. I eventually realized that if you were performing similar operations multiple times, putting all of this information into an object where the details are abstracted away from the user probably reduces errors. But I didn't understand that until I had gone away and thought about it a lot. Please rewrite the beginning of the paper to emphasize this and any other advantages of classes that I may have missed.

We hope that this has been clarified by the restructured introduction.

Perhaps this is the same point, but I was confused by the sentence "By providing a middle layer into the workflow, it is much simpler to apply the analysis technique to multiple data sources, to share it with others and to expand upon it in the future." I do not understand what "providing a middle layer to the workflow" means, and I would like to understand more about why classes were chosen for this task.

This sentence has been removed as part of the restructure.

2. I can't find any examples of this python package being used on gridded datasets that are not based on NEMO. It is fine if this package (and hence framework) is actually mainly designed for NEMO data, but then the paper should clearly state this. If this package will be applied to other gridded datasets, I'd like to see a discussion of how different kinds of data (netcdf, zarr, binary) could be read by the package, e.g. via xarray. Lines 30-37 say some really important things about the need for lazy loading, but it's no good having lazy loading if I have to rewrite all my data in a different format in order to even load it into the package.

The package is not only designed for NEMO and the Gridded class is intended to be a generic class for any model data. However, the reviewer is correct that it has only been applied in earnest to NEMO model output at this time.

In addition, I am not sure that this package makes full use of lazy loading. If the data is in netcdf format with no use of kerchunk indices, then you must load the whole netcdf file in order to access the data. The dask tutorial on the COAsT website is a bit lacking here. Certainly computation can be delayed and some parallelization should be possible because the objects are based on xarray, but again it's not clear why building these new classes is helpful, because the user has to use xarray/dask in order to parallelize anyway. Why not just use xarray objects directly?

Fundamentally, the standardized structure described in the paper is separate from lazy loading. Using xarray and dask allows us (or the user) to make use of lazy loading if they wish. Some of the analysis routines within COAsT do use lazy loading, however others don't at the time of writing. The paper is meant to be an overview of the ideas underlying COAsT (anticipating future updates), rather than of specific routines.

3. It seems to me that COAsT is a bit of an "everything but the kitchen sink" package at the moment. Having an expandable code base is nice, but xarray already exists and some more clarity on the goals of COAsT would certainly help people to understand what is going on here.

Hopefully the restructured Sections 1 and 2 help with this.

4. If this manuscript is meant to document the python package (and the first half of the manuscript suggests that it is), then I'd like to see a significant discussion of testing. Part of having an expandable code base is having well-designed tests. I see the package has some testing set up. Good code coverage is also necessary for the testing, so that untested code isn't constantly being added.

COAsT indeed has unit_testing and good coverage (67%). Testing is added to the Exandible Codebase principle. In particular:

*"However, having a mechanism for unit testing old and new contributions (with good coverage) is fundamental to maintaining a working codebase. A system of testing is an essential tool for rapid error trapping, when creating a robust codebase by multiple authors, against a backdrop of evolving module dependencies"*

5. I like the Matched Harmonic Analysis section, but I'd like to see a bit more context at the beginning. What is the overall goal of the comparison?

We have modified the first two paragraphs of Section-3.1 to add some additional context for why the Matched Harmonic Analysis is necessary.

6. I was not able to understand the description of CRPS provided between line 290 and 299. Please provide more detail on what F x and y represent.

We have added some more explanation:

"The CDF $F$ is derived from all model values from within some predefined radius around the observation $x$, as described above. $F(y)$ is then the value of this CDF for a single element of the model radial dataset. More intuitively, the CRPS can be thought of as the mean square error between modelled and observed CDFs."

7. The code actually used to make the figures presented here does not appear to be available anywhere (potentially it is located somewhere in the package, but its location is not given). For a paper that talks about reproducibility, I think that the plotting code should at least be provided. Ideally the datasets used to generate the figures would also be made available, but I understand that they might be too large.

Unfortunately, in this case the data is too large to share easily with the reader. The code used to plot is now included in a separate repository, provided in the code availability statement section of the paper.

Minor points:

1. I'd like to see some citations for technical concepts like lazy loading, chunking etc. I know that traditional references for these concepts may not be available, but I think non-expert readers would benefit from some references.

We have been unable to find specific/traditional references for lazy loading and chunking, but agree that some pointers would be useful to some readers. We have added the following non-traditional references.

https://en.wikipedia.org/wiki/Lazy_loading [last accessed 5 Apr 2023]

https://docs.dask.org/en/stable/array-chunks.html [last accessed 5 Apr 2023]

2. I would also like to see more citations for concepts introduced between line 145 and line 165. e.g. "the estimation of tides is a vital step for the validation of sea surface height in our regional models" Please reference an example. "Non-tidal residual signals can be generated by many processes but in coastal regions the modest (sic) significant are generated by atmospheric processes". How do we know this?

We have added the Sea Level Science textbook (Pugh & Woodworth) as a citation.

3. Figures 1 and 3 have colorbars with white in the middle. I would recommend choosing a different colorbar so that we can see all the observations.

We agree that the white in this colormaps may obscure some data. However, in this applicatin, we specifically wish to draw attention to where the model does not perform so well, as this is what we are seeking to improve.

Typos:

1. Line175: "quick and easy" should be "quickly and easily"

Fixed.

**REVIEWER 2**

**General comments**

The authors propose the foundation of a framework based on the COAsT Python package to evaluate kilometric scale regional modeling outputs against observations. After describing their principles to construct such a workflow, they showcase two applications: the comparisons to tide gauges and mooring profiles along the coasts of the Northwest European Shelf.

The paper is well written, and the balance between the description of the method and the two applications is good. The authors claim that validation is part of the model's development and that analyses should be automated, and they are correct. Their work is a cornerstone for achieving such a goal.

The main issues concern, first, the position of their tool among all the python packages dedicated to ocean analyses, and second, the fact that some of their fundamental principles (scaleability, independence to data source) may not be fully demonstrated.

**specific comments**

Title: As the framework is entirely related to the COAsT Python package, why not mention COAst in the title?

We have renamed the paper, which now reads:

"Using the COAsT Python Package to Develop a Standardised Validation Workflow for Ocean Physics Models"

**Introduction**

The authors mention that they use the COAsT Python package in their standardized validation framework. But throughout the paper, they describe classes, methods, and analyses available in COAsT. This paper looks like a scientific/engineering application of the COAsT library. Thus, the capabilities and novelty of the COAsT package should be well explained in a specific paragraph to outline its contribution among other ocean analysis tools. And the authors must include references to other related packages (in Python language, at least) in the paper.

We have restructured the opening sections (Introduction, COAsT Framework) of the paper to make it clear what the intent of the paper is. This means more of an emphasis on COAsT being the fundamental aspect of the paper.

L66. The authors specify "many" principles can be satisfied by using COAsT package. Indeed, principles 3,4,5 are straightforward but not 1 and 2. So it is worth clarifying which principles still need to be fully achieved and why.

We hope this is clearer with the restructured introduction.

**Methods**

L116. Are there any intermediate steps, such as saving sub-datasets in zarr format?

There are intermediate steps. The validation against EN4 temperature and salinity profile data breaks the process into preprocessing, processing and postprocessing steps. At present the intermediate files are stored in netCDF format. But this is an evolving package and we have interest in looking at alternative options and efficiencies.

L127. Please, for easy reading, summarize what differs between the two configurations (even though one can get the information from the table). And clearly state what is rigorously the same.

It is hard to pin down exactly what are the differences and similarities in the NEMO versions between CO7 (NEMOv3.6) and CO9p0 (NEMOv4.0.4) as there are structural changes and a number of bug fixes. One of the structural changes abstracts the computation of the grid to now be a preprocessing step. But the code to achieve this is intended to be the same and the supplied bathymetry is the same. Other structural changes which we have tried to match include:

Bulk forcing implementation: We used the "NCAR" algorithm (Large and Yeager 2008) in CO9p0 in an attempt to closely match the CORE bulk forcing algorithm (Large and Yeager, 2009) in CO7

Lateral diffusion of tracers: We attempt to replicate the constant value used in CO7.

Lateral diffusion of momentum: This _is different_ between model runs. For stability purposes we deviate from a constant value (as used in CO7) and use the NEMO 4 option that varies diffusion according to grid scaling and local velocity.

Tracer advection: In CO7 the Total Variance Dissipation (TVD) was used. In NEMO 4 the closest equivalent is the Flux Corrected Transport scheme (FCT), which is set to 2nd order in horizontal and vertical directions.

This has been added to the discussion following Table 3.

Table 3. What is the coastline product? Also, information on the output files is helpful to understand better the difficulty of reading (more complicated and long as the number of files is large even with xarray reading methods) and the impact of temporal interpolations.

The model outputs SSH at hourly frequency (tidegauge data typically reports at 15 minute intervals). The model outputs 3D temporal mean T/S at daily frequency. These are added to the table.

There is no specific "coastline product". The coastline in the hydrodynamic model is determined by the bathymetry dataset (Graham et al., 2017). The locations from the tidegauge dataset are determined at much greater accuracy when the sites are leveled. These metadata are in the the GESLA dataset. The model and tidegauge dataset are paired by taking the nearest wet simulated grid box to the tidegauge location.

L131. Please indicate the multiple sources.

These harmonics are derived from many timeseries of different lengths, origin and sources. They come from tide gauges, bottom pressure sensors, moorings etc. The point here is that these observations come from many sources, and their varying lengths and properties means that the harmonic analyses which have significant uncertainty. So using such an eclectic datasets allows us to  apply our harmonic uncertainty to the analysis. We have added to the paper:

"These eclectic sources include tide gauges, moorings and bottom pressure sensors with a very large range in analysis length. Using such a variable dataset allows us to test the harmonic matching and uncertainty discussed in Section-3"

L132. "These locations... Section-3" should be removed to stick to the general description. But a (bigger) figure dedicated to the locations and types of data is welcome because it is much easier to see the locations of the tidal amplitudes and phases in Figure 3. And it will allow for renumbering the figures in a progressive manner in accordance with the text: current 1, 2, 3, 4 numbers will become 4, 3,2, 5. Moreover, the regions over which profile comparisons are averaged could be visualized.

We have removed the reference to Figure-3 in this section and postponed it to Section 3.

We have added a new figure showing the regions used for analysis.

**Validation against tide gauge data**

L182 Reformulate? For each location, the analysis lengths have been identified from observations.

Agreed. Changed.

L194. "it must not be ignored". What do you mean? The harmonic analysis can be performed, but considering the uncertainty?

We have reworded this line:

"Using an MHA can reduce this uncertainty, but where this is not possible the uncertainty must be considered instead"

L195 - L203 - L228. The reader can get confused. L195, it is precised "both the MHA approach and an application of the harmonic variability". But L203 is "As discussed in Section-3.1, we cannot apply a matched harmonic analysis to this analysis". L228 "apply the matched harmonic analysis described in Section-3.1". Could you add in the sentence L195 (section 3.2) and (section 3.3)?

The confusion arose because not all of the observational data is of sufficient quality to permit the Matched Harmonic Analysis with the simulated data. Nevertheless we want to maximise the value of all the historical data, so rather than discard it we instead devise a method to compute uncertainty around these data. The following changes have been made to make this clearer:

- Clarification that raw timeseries data is missing for some observations in 2.2.2 (Observational Data description)
- Introduction of a subsection "3.1.1 Harmonic uncertainty estimation when MHA is not possible"
- In Section-3.2: Better signposting of the workflow (3.1.1) when MHA is not possible
- In Section-3.3: Better signposting that the data is filtered to only use those suitable for MHA in the calculation of non-tidal residuals.

L258. And we can conclude that the models do not capture large events, or that large events are underestimated in the models, right? A short interpretation of the figure is welcome.

We have added a couple of new sentences to highlight this interpretation:

"However the figures also show that both configurations are underestimating large events, and the larger the observed event gets, the more their number is underestimated. It also seems that neither model is sufficiently sensitive to atmospheric forcing and that coastal effects, such as resonance, are not being adequately represented."

Validation against profile data

L285-... The separation between the methodology and the results makes the reading a bit confusing. Please, move this sentence into the previous paragraph and add a reference to figure 7. Then start a new paragraph about CRPS at the sea surface.

Done.

L304 for temperature and salinity.

Added:

The absolute error metric allows us to see how each model performs, **for temperature and salinity**, at different depths and in each region.

L306. Please comment on the results shown in figures 9-10.

We have added the following sentences:

"In this case, we see similar CRPS values across all radii. For SST, the CO7 configuration performs best (i.e. smallest values) for most regions. The opposite is true for SSS. For both variables, the Norwegian Trench and Kattegat generally have the highest (worst) CRPS values."

Figures 7-8 and 9-10. Why do the regions differ between the two panels? "Irish sea" versus "off Shelf"?

These figures have now been updated. Regions are now consistent between figures.

Furthermore, we have added a map of regions in Fig 7.

Further discussion

L314 Scaleability is one of the fundamental principles on which this tool is based, and rightly so. However, the showcases do not fully demonstrate this capability. The comparisons use a large number of vertical profiles and time series. In this sense, scalability is achieved. Even though the model outputs are huge, the analyzed data volume (time series and profiles) remains small because Pangeo tools (dask, xarray, etc.) effectively sub-select specific locations. Perhaps the authors should slightly moderate their conclusion or clarify the potential beyond the actual results shown in the paper. Similarly, the design intends the data source independence, but so far, only NEMO model outputs seem to have been used.

Regarding scalabilty, this application is considered large by the standards set a few years ago when the regional model demonstrated here had a significant increase in resolution: Old workflows running matlab on desktops are no longer sufficient. We concede that we have not thoroughly tested its 'scalability' instead pragmatically developing and using it for

our applications of interest. Nevertheless we are now starting to deploy the T/S validation to a global ¼ and then 1/12 deg simulation. While we wrote the COAsT package and scripts mindful of these computationally more demanding downstream applications, we also always expected to refine it as the tasks required.

It is true that only NEMO has been used for this paper. However, the classes within COAsT are generic and aim to be applicable to any regularly gridded model data.

We have toned down the language regarding scalability. *"The principles introduced aim to ensure that assessments are scaleable…"*

We have also clarified the COAsT success is largely a function of its components: "*The package, along with its components (especially Xarray), are important tools, allowing us to adhere to the assessment principles with ease.*"

**technical corrections**

Table 1. Should be Gridded (t_dim, z_dim, y_dim, x_dim)? Is the order correct for Indexed?

You're correct, y_dim should be before x_dim. This has been changed.

Table 2. Profile: isn't time a coordinate?

It is, thank you. We have corrected it in the table.

Table 3. Initial conditions "Analysis period starts 2004" should be set in the text.

Fixed. In Section-2.1.

L201. Figures 1 and 4. Figures 2 and 3 are excluded.

Fixed.

L214. Just curious. From figures 1 and 4, CO9p0 improves the representation of the tides along the NW coast of the UK? Why? Different coastline? Or bottom friction formulation?... Even though I regret it, no additions should be made to the paper to explain the improvements, as it is not the topic of this paper. So feel free to answer my question or not.

The amplitudes are improved along the NE coastline. In some ways this is a bit disappointing as the code bases were supposed to be as close to a version bump as we could manage. However, given that this change appears one would reasonably ask why? Our interpretation is that the newer codebase is doing a better job of the tidal Kelvin wave propagation southward along that coastline because of the changes in the implementation of momentum diffusivity. On the other hand, it could also be a shift in the amphidrome in the North Sea. One of the useful things this analysis did was highlight that pretty much all of the offshore observational data in the North Sea is no longer of sufficiently high quality to be

used to adjudicate on model performance. Until we computed the uncertainties in the observational harmonics we used to get distracted by the errors they presented.

L244. Correct the sentence "Both models have are similar"

Fixed

Figure 2. Please complete the legend. Could the colorbar on the right panel be changed? It isn't easy to distinguish the squares. And why not reduce the geographical extension of the domain?

The domain was chosen to be consistent with other figures. As with Fig 1 and Fig 3, the colormap was chosen to highlight the disparities, so small values appear less consequential than larger values.

Figure3. Please (a) add the units (day?) (b) according to the text, the unit is % of the M2 amplitude (not m).

Fixed.

Figure 5. The color scale is not discriminating in panels a and b.

It is difficult to choose a colorscale that discriminates as values are very close generally. The third panel in the figure is meant to help the reader discern these differences.

---

## Author Response (AR2)

We are grateful for the reviewer's careful reading and comments which have served to sharpen the manuscript. Below we respond to the comments point by point.

**Reviewer 1**

1. There are a few issues with the order in which things are mentioned.

- Lines 29-34 do not mention the sections in the right order and one of the section numbers isn't rendered.

We have fixed the ordering and broken cross reference.

- It is customary to put the tables and figures in the same order as you refer to them in the paper, and this isn't how they are ordered at the moment.

We have reordered the figures.

- Lines 125-130 are mostly a repeat of lines 29-34.

Moved lines to 29-34 from 125-130

- Figure 4 does not seem to be mentioned in the text.

All figures are now mentioned in the text.

2. I suggest citing a couple of other packages for comparing models with observations, e.g.

- Castruccio F. S., 2021: NCAR/metric: metric v0.1. doi/10.5281/zenodo.4708277

- Roberts, C.D., 2017: cdr30/RapidMoc: RapidMoc v1.0.1. doi:10.5281/zenodo.1036387

We have integrated these citations, along with some other packages in a revised Introduction section. Please see lines 24 - 37.

3. Line 85: "By pairing it with dask" What does "it" refer to? Please rephrase.

Sentence rewritten:

" By integrating Dask and Xarray into COAsT, the user has access to a powerful system that provides lazy loading, chunking and parallel code. "

4. Line 86: "the user also has access to data analysis concepts". I'd argue that these are tools or applications. I have access to concepts without the package because concepts are entirely cerebral.

Replaced 'data analysis concepts' with 'data analysis tools'

5. I found the description of CRPS to be better, but it would still be helpful to clarify: is the integral over y an integral over time, space, or both?

Added an extra sentence:

"The difference is integrated spatially over data from the radial neighbourhood."

6. I'd love one more sentence about what the "mask_maker" class is.

Added an extra sentence (line 338):

This class contains predefined region definitions that help standardize spatially aggregated metrics.

7. Could the caption of Figure 3 please describe the meaning of the legend in panel a?

Added clarification that legend is in days. (Now Figure 1a)

Typos
1. Lines 106: "re used" You do not need a space
2. Line 113: eliminate "However"

Fixed both typos.

**Reviewer 2**

First, I would like to thank the authors for considering most of our comments and suggestions and for their frank answers. I regret, however, that nothing was added in the introduction to highlight how the package fits among the many python packages dedicated to ocean analyses.

We have added a paragraph to the introduction, and rewritten some other parts of the introduction to provide this context. Please see lines 24 - 37.

Nevertheless, given the useful philosophy developed throughout the article and the valuable validation methods, the paper is worth publishing.

A few (~12) corrections and improvements are still requested and are listed below:

L31 Remove (see Section-??) and join the two sentences. "The Gridded class is used to read, represent and manipulate output from two NEMO model runs, and its use interactively with the Profile and Tidegauge classes allows a comparison between the model and observed data.

Missing cross reference has been fixed.

Sentence replaced as suggested.

L34. Rewording: Information on the observation and modeled data used are given in Section 2.

L34-36 Please delete the paragraph, it is a conclusion, not an introduction. Also, "previous section" is non correct.

Deleted and reworded.

The sentence L34 "The package is open source and all of the code is freely available via Github (www.github.com)" could be replaced after L352.

We remove this line from the introduction and better describe the package availability in the Code Availability Statement.

The sentence "Using COAsT provides a level of transparency which aids knowledge sharing and discussion" could be moved to L366 after slight rewording.

This sentence has been removed.

L48 Even though the package aims to be extended to other ocean numerical models, it is not yet the case (see the documentation, the answer to reviewer 1, and the fact that analyses shown are based on NEMO only). Experiences always prove that despite the best prerequisites, extensions are not that straightforward. So please remove the reference to ROMS or at least add "in the future".

Reworded these sentences to read:

Typically, this kind of data would come from the output of a numerical model. At the time of writing, the package has been tested and used with output from the NEMO model, although this could be extended to other models in the future. The data should be stored in any xarray compatible file (e.g. netCDF, zarr).

L67-L68 The sentence "It labels these dimensions with 1-dimensional time coordinates, 2-dimensional latitude and longitude coordinates and 3-dimensional depth coordinates." is unnecessary (repetition of L47). Please remove it.

Removed.

Moreover, the dimensions do not fully comply to the C-Arakawa grids.

This is true within a single instance of the class. When using related grids with offsets, an instance of the class can be made for each grid. This is not mentioned in the paper as we do not consider variables on these other grided.

L71. Could you please add a few words about the analysis classes? I am not sure whether the tidal and profile analyses are directly available from the COAsT package. It seems they are saved apart under https://github.com/JMMP-Group/NEMO_validation.

or

Move paragraph L65-L71 ("At the time... Table-2) to L62, just before "Analysis classes...

Additional clarity on the COAsT analysis classes is added  (L72): "For example, analysis classes include *GriddedStratification*, *ProfileAnalysis*, *TidegaugeAnalysis*, *Transect* and *Contour*". Furthermore improved tracability of the code used, including the COAsT and NEMO_validation repositories, is provided in Code Availability Statement section.

L82-L87 add no more information. Please consider just removing them ("All three of ... chunking and parallel computation")

Removed.

L116 The three previous sentences are general and do not give any indication of what has been done in COAsT. After "module dependencies", consider adding a sentence stating that in COAsT, unit tests cover 67% of the package.

We have added a sentence:

For COAsT, unit tests cover approximately 67\% of the package at the time of writing.

Section 3. The numbering is unbalanced. Why not numbering L180 as follows?:

3.1 Harmonic Analysis
3.1.1 MHA
3.1.2 Harmonic uncertainty estimation where MHA is not possible (L203)

We have changed the section titles as recommended.

L229-230 Please clarify.
- Was MHA applied at all locations where it was possible and basic HA everywhere else?
- Was harmonic uncertainty estimated at every location (even where MHA was possible) and then used to "mask" correct assessments?

For the analysis presented in Figures 2 and 3, a normal harmonic analysis was applied everywhere, and uncertainty applied to 'mask' small differences between the models. We have added to the text to clarify this:

"Here we perform a normal harmonic analysis at each point using all available model data. Then, where differences between the model and the observations are smaller than the uncertainty, they are deemed insignificant and coloured grey in the figures. "

L237 In the figureS

Fixed.

Figure 1. Again the numbering is confusing because Figure 1 is detailed after Figures 2 and 3.

Figure ordering has been fixed.

Figure1/4 description. Please specify which method (MHA or/and just HA) has been applied. (see comment L229-230)

We have edited the captions of Figures 2,3 and 6 to add more information on harmonic method used.